# LoRA-Gen: Specializing Language Model via Online LoRA Generation

## Abstract

Recent advances have highlighted the benefits of scaling language models to enhance performance across a wide range of NLP tasks. However, these approaches still face limitations in effectiveness and efficiency when applied to domain-specific tasks, particularly for small edge-side models. We propose the LoRA-Gen framework, which utilizes a large cloud-side model to generate LoRA parameters for edge-side models based on task descriptions. By employing the reparameterization technique, we merge the LoRA parameters into the edge-side model to achieve flexible specialization. Our method facilitates knowledge transfer between models while significantly improving the inference efficiency of the specialized model by reducing the input context length. Extensive experiments show that LoRA-Gen outperforms the conventional LoRA fine-tuning, which achieves competitive accuracy and a 2.1x speedup with TinyLLaMA-1.1B on common-sense reasoning tasks. Besides, our method delivers a compress ratio of 10.1x with Gemma-2B on intelligent agent tasks.

## 1 Introduction

The principle of scaling laws (Kaplan et al., 2020) demonstrates that increasing the size of Large Language Models (LLMs) can significantly improve cross-task generalization. However, due to the constraints of their enormous size, generic LLMs struggle to achieve a good balance between efficiency and effectiveness when addressing domain-specific tasks or preferences. Accordingly, improving smaller edge-side language models which are deployed on edge devices to run locally with a compact size in specific tasks (Fu et al., 2023; Grangier et al., 2024; Shen et al., 2024) is receiving increasing attention in both academic research and industrial applications. Many approaches utilize parameter-efficient fine-tuning techniques (Houlsby et al., 2019; Li & Liang, 2021; Lester et al., 2021; Hu et al., 2021), particularly LoRA (Hu et al., 2021), to train on specific datasets for specialization. However, this method may encounter the issue of catastrophic forgetting, which can result in a decrease in performance on other unseen tasks (Feng et al., 2024; Huang et al., 2023a).

To alleviate knowledge forgetting in specialized training, recent approaches (Dou et al., 2024; Gao et al., 2024a; Yang et al., 2024c; Li et al., 2024a) leverage the flexibility of the Mixture of Experts (MoE) for LoRA training. Specifically, as shown in Figure 2(b), they integrate a group of multiple LoRA components as experts within the language model, allowing the language model to control the selection of LoRA components during token generation. However, these methods introduce additional inference costs due to the extra experts and control units. LoRAHub (Huang et al., 2023b), on the other hand, pre-trains a set of task-specific LoRA components and employs a manually designed parameter-free optimization method for selection. Nevertheless, the effectiveness of above mentioned approaches is limited by their model scale, resulting in constrained performance and generalization capabilities on unseen tasks. Therefore, this paper explores a new perspective: *utilizing a large cloud-side model to generate parameters for a smaller edge-side model to achieve better specialization.*

To achieve it, we propose a new LoRA generation framework, termed LoRA-Gen. As shown in Figure 2(c), our method can be divided into two parts: Online LoRA generation and Specialized LM. The former is used to generate LoRA parameters based on the task-defined system prompt, while the latter facilitates efficient batch inference for user input. Specifically, a fine-tuned large language model, LLaMA3 (Dubey et al., 2024) and a mixture of LoRA experts are deployed in the cloud.

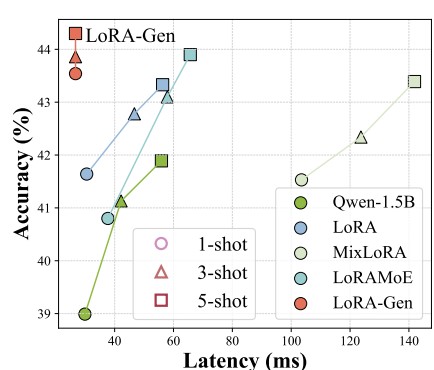

**(a) Vanilla LoRA Paradigm**   **(b) LoRA-MoE Paradigm**   **(c) LoRA-Gen for Specialized LM**

Figure 2: Comparison of different LoRA-based fine-tuning strategies.(a) Vanilla LoRA is fine-tuned on the target task and then merged into the source model. (b) LoRA-MoE introduces additional LoRA experts to improve the generalization performance. (c) Our LoRA-Gen presents an online task-specific LoRA generation framework that customizes a specialized LM for edge-side users.

The cloud-side language model generates a set of meta tokens based on the given system prompt. Each meta token corresponds to a transformer layer in the edge-side language model, utilizing these tokens to control the composition of parameters from the LoRA experts. Similarly to vanilla LoRA, the combined parameters are further merged into the edge-side LM through reparameterization, resulting in an efficient specialized model.

As shown in Table 1, our LoRA-Gen offers four advantages over previous methods: i) Context compression for unseen tasks: LoRA-Gen dynamically compresses the task-specific system prompt (*e.g.*, task descriptions, few-shot samples, and chat templates) into the LoRA weights, which significantly reduce the context length for the specialized models. ii) Reparameterized model: Unlike LoRA-MoE (Dou et al., 2024), our approach employs reparameterization techniques to merge the generated LoRA weights into the original parameters, thereby avoiding additional inference costs. iii) Inference-time specializing: Our method does not require any additional training, including few-shot tuning when specializing the model for unseen tasks. It only necessitates a single-turn inference on the system prompts to obtain the specialized model parameters, which simplifies model deployment. iv) Knowledge Transfer: LoRA-Gen allows the cloud side and edge side to utilize different models, enabling the injection of knowledge from the large cloud model into the edge model through reparameterization, which enhances performance effectively as shown in Figure 1.

Figure 1: Accuracy-latency curves comparison with various few-shot numbers on ARC-c task. Best view in color. Base model is Qwen-1.5B.

We conduct extensive experiments to validate the effectiveness of LoRA-Gen on various commonsense reasoning tasks as well as an agent benchmark. The results demonstrate that our method balances both performance and efficiency, showing significant advantages across eight language datasets. For the edge-side model of TinyLLaMA-1.1B, LoRA-Gen outperforms vanilla LoRA fine-tuning by a remarkable margin with only 1/6 sequence length, +1.3% on harmonic mean, and 2.1x speedup. Moreover, for the Gemma-2B model, LoRA-Gen demonstrates competitive performance on unseen agent tasks. Additionally, since it does not require the input of agent definitions during inference, it achieves a remarkable 10.1x compression ratio.

## 2 RELATED WORK

### 2.1 PARAMETER-EFFICIENT FINE-TUNING

Given the billions of parameters in LLMs and the limitations of current hardware, fully fine-tuning LLMs in the traditional manner is often impractical. To address this, several parameter-efficient

| Method | Context Compression for Unseen Tasks | Reparameterized Model | Inference-time Specializing | Knowledge Transfer |
|---|---|---|---|---|
| ICL (Dong et al., 2022) | ✗ | ✗ | ✔ | ✗ |
| LoRA (Hu et al., 2021) | ✗ | ✔ | ✗ | ✗ |
| LoRA-MoE (Dou et al., 2024) | ✗ | ✗ | ✔ | ✗ |
| LoraHub (Huang et al., 2023b) | ✗ | ✔ | ✗ | ✗ |
| LoRA-Gen | ✔ | ✔ | ✔ | ✔ |

Table 1: Characteristics comparison with other counterparts. ICL indicates the in-context learning.

fine-tuning (PEFT) methods have been developed. Adapter-based approaches (Mahabadi et al., 2021; Zhou et al., 2024; Zhang et al., 2024b) involve inserting trainable adapter layers into various blocks of pre-trained models. Soft prompt methods (Li & Liang, 2021; Liu et al., 2022) adjust a small trainable prefix vector to adapt LLMs to new tasks. Unlike these methods, LoRA (Hu et al., 2021) minimizes the number of trainable parameters for downstream tasks by freezing the pre-trained models and tuning only additional rank decomposition layers. This method approximates weight adjustments during fine-tuning without incurring extra costs during inference. Building on this, AdaLoRA (Zhang et al., 2023) dynamically adjusts the parameter budget among weight matrices, while DoRA (Liu et al., 2024b) fine-tunes both the magnitude and directional components decomposed from pre-trained weights. VeRA (Kopiczko et al., 2024) further reduces the number of trainable parameters by utilizing shared low-rank layers and learnable scaling vectors.

## 2.2 LoRA Meets Mixture of Experts

Leveraging its lightweight nature, LoRA is utilized in Mixture of Experts (MoE) architectures to enhance performance. MoLoRA (Zadouri et al., 2023) incorporates LoRA adapters as experts on top of pre-trained models and uses a router layer to integrate these experts. MOELoRA (Liu et al., 2024a) applies this framework to various medical domain tasks, though it requires task type input for the router. LoRAMoE (Dou et al., 2024) introduces multiple LoRA experts into the feed-forward block to mitigate knowledge forgetting during the instruction-tuning phase. LoraHub (Huang et al., 2023b) allows a dynamic assembling of LoRA modules on various tasks and even unseen tasks by combining adapted LoRA modules. Additionally, MoLA (Gao et al., 2024a) proposes layer-specific experts, allocating a varying number of LoRA experts to different layers to boost performance.

## 2.3 Context Compression

With the rise of in-context learning (Wei et al., 2022) and agentic pipelines (Yang et al., 2024b), LLMs often need to process thousands of tokens, potentially exceeding their maximum context length. Unlike methods that extend the context window of LLMs, context compression offers an efficient way to reduce the input prompt length. There are two primary methods of context compression: hard prompt and soft prompt. Selective-Context (Li, 2023) and (Jiang et al., 2023) exemplify hard prompt methods by removing low-information content at the lexical level (e.g., sentences, words, or tokens) to shorten the prompt. On the other hand, gisting (Mu et al., 2023), AutoCompressors (Chevalier et al., 2023), ICAE (Ge et al., 2024), and 500xCompressor (Li et al., 2024b) represent soft prompt methods that compress input prompts into a small number of special tokens. In contrast to these approaches, we propose compressing the context into rank-decomposition layers using LoRA methods.

## 3 Methodology

In this section, we first review the LoRA-based Mixture of Experts fine-tuning paradigm and then elaborate on our LoRA-Gen, which generates task-specific LoRA weights according to the system prompt for edge-side language models.

### 3.1 REVISITING MIXTURE OF LoRA EXPERTS

LoRA (Hu et al., 2021) improves the efficiency of fine-tuning by significantly reducing the number of trainable parameters. Formally, it updates the weight matrix $W \in \mathbb{R}^{d' \times d''}$ by using a low-rank approximation via two decomposition matrices $A \in \mathbb{R}^{d' \times r}$ and $B \in \mathbb{R}^{r \times d''}$ with a low rank $r$ ($r \ll min(d', d'')$) as follow:

$$\widetilde{W} = W + AB. \tag{1}$$

Trainable low-rank decomposition matrices can capture the underlying patterns of downstream tasks under the guidance of the task-specific direction (Hu et al., 2021). Moreover, another effective approach, the Mixture of Experts (Jacobs et al., 1991; Jordan & Jacobs, 1994) termed MoE, treats multiple networks as experts and seeks to take advantage of their strengths in a hybrid framework. This method aims to combine the advantages of different models, resulting in improved generalization and overall performance. Typically, a MoE layer consists of $n$ experts, denoted as $\{E_i\}_{i=1}^n$ with a router $R$ as the gate for expert allocation. Given hidden states $\{h_j\}_{j=1}^s$ of a sequence with the length of $s$, the output of the MoE can be formulated as:

$$h'_j = \sum_{i=1}^n R_i(h_j) E_i(h_j) \tag{2}$$

Considering the efficiency of LoRA and the strong performance of MoE, (Li et al., 2024a; Dou et al., 2024; Gao et al., 2024a; Yang et al., 2024c) integrate LoRA into the MoE plugin, boosting the fine-tuning performance by utilizing the mixture of LoRA experts, effectively blending the strengths of both methods.

### 3.2 ONLINE LoRA GENERATION

**Overview.** The mixture of LoRA experts has showcased reasonable performance in fine-tuning for specific tasks. However, there remains a gap in its effectiveness for multi-task learning and the generalization to unseen tasks. Additionally, most LoRA-MoE (Li et al., 2024a; Dou et al., 2024) methods require calculating the expert routing for each token individually, which significantly increases the computational complexity. To address these challenges, we propose a new framework, termed LoRA-Gen that generates task-aware LoRA via an online large language model with system prompts (including few-shot samples, task description, role specification, and the conversation format) as presented in Figure 3. In the following, we elaborate on our LoRA generation method and the reparameterization of the edge-side language model.

**Cloud-side LM & Meta Token.** In adherence to meta-learning (Hospedales et al., 2021; Finn et al., 2017), we construct a unified representation of the task-related information to achieve generalization capabilities for various tasks, relying on cloud-side language models to facilitate this process. Specifically, Given a series of few-shot samples or task-specific system prompts, the cloud-side LM appends $L$ special tokens $\langle meta \rangle$ behind them and transfers the inherent knowledge into these tokens with causal masks in a single forward pass. We define these tokens as meta tokens $\{T_i^{meta}\}_{i=1}^L$, where $L$ represents the number of layers of the subsequent edge-side language model. Each meta token is associated with a transformer layer in the edge-side LM.

**LoRA Expert Pool.** Our initial attempt is to generate LoRA parameters directly through a continuous projection on the meta token. However, the expansive parameter space poses optimization challenges, making the model susceptible to overfitting and hindering generalization, whose analysis refers to Table 8. Therefore, similar to the previous works (Dou et al., 2024), we adopt an alternative solution by introducing the discrete MoE mechanism. Specifically, as shown in Figure 3, we construct a LoRA expert pool of $n$ experts, whose weights are defined as $\{E_i\}_{i=1}^n$. Each LoRA expert contains three LoRA blocks, corresponding to the gate linear layer, up linear layer, and down linear layer in FFN of the edge-side model, respectively. Different from the LoRAHub (Huang et al., 2023b), these experts are trained in an end-to-end manner.

**Routing Module.** To control the composition of experts, we propose a routing module using meta tokens. Unlike the token-wise LoRA-MoE (Dou et al., 2024), our MoE is layer-wise. We apply an individual MoE for each transformer layer in the edge-side LM, and all tokens in a sequence use

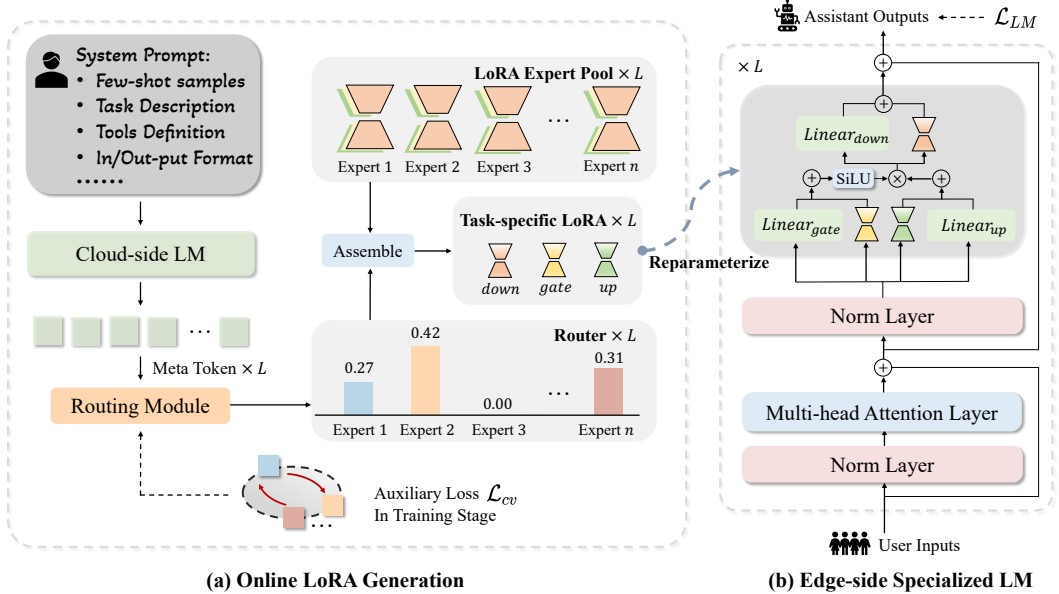

**(a) Online LoRA Generation**      **(b) Edge-side Specialized LM**

Figure 3: Overview of our proposed LoRA-Gen. Given the system prompts by users, a large language model first generates meta tokens in a single forward pass. With a routing module, we obtain the gates of all experts in the online LoRA pool. After assembling, we produce the specialized LoRA on the cloud side and deploy it to the edge-side language model by merging the LoRA weights.

the same composition. For simplicity, the routing module consists of two linear projections with a Batch Normalization (BN) layer. Incorporating a BN layer can further increase the diversity of router output, promoting the utilization of a wider range of experts. In formal, the router $R^i \in \mathbb{R}^n$ of $i$-th layer of edge-side LM can be formulated as:

$$R^i = BN(f_2 \circ \varsigma \circ f_1(T_i^{meta})), \tag{3}$$

where $f_1$, $f_2$ are the linear functions and $\varsigma$ denotes the $SiLU$ activation function. We attempt to increase selection randomness and balance expert loads, we apply Gumbel-Softmax (Jang et al., 2016), which can be formulated as:

$$\text{Gumbel-Softmax}(R_t^i) = \frac{e^{R_t^i + g}}{\sum_{j=1}^n e^{R_j^i + g}}, \quad \mathbb{E}_{g \sim \text{Gumbel}(0,1)}(g) = 0. \tag{4}$$

Nevertheless, the Gumbel-Softmax strategy shows a significant reduction in generalization performance, which is reported in experiments of Section 4.4, To this end, following previous methods (Li et al., 2024a; Dou et al., 2024), we adopt a KeepTOP-K strategy to select experts in a deterministic manner:

$$\widetilde{R_t^i} = \frac{e^{R_t^i}}{\sum_{j=1}^n e^{R_j^i}}, \quad \text{TOP-K}(\widetilde{R^i}) = \{\overline{R_t^i}\}_{t=1}^K, \quad G_t^i = \begin{cases} \frac{\widetilde{R_t^i}}{\sum_{j=1}^K \overline{R_j^i}} & \widetilde{R_t^i} \in \text{TOP-K} \\ 0 & \text{else} \end{cases}, \tag{5}$$

where $G_t^i$ represents the the gate of $t$-th experts for $i$-th decoder layer of the edge-side language model. Consequently, we generate task-specific LoRA weights as follows:

$$\text{LoRA-Gen}^i = \sum_{j=1}^n G^i E_j. \tag{6}$$

**Reparametrization.** Similar to LoRA, we use the reparameterization strategy to merge the generated LoRA parameters into the FFN layers of the edge-side model. In contrast to the LoRA-MoE, our method is cost-free during inference, which needs no additional components in the specialized edge-side LM.

## 3.3 TRAINING TARGET

**Auxiliary Loss.** Balanced load of MoE structure is essential for capability of generalization and stability (Jacobs et al., 1991). Without constraints, the routing module tends to select a fixed small set of experts, leaving other experts unused and causing load imbalance. To mitigate this issue, we introduce a soft constraint with the coefficient of variation as the auxiliary loss, encouraging a more balanced usage of the available experts. Formally, the constraint can be formulated as:

$$\mathcal{L}_{cv} = \alpha(\frac{\sigma(G)}{\mu(G)})^2,\tag{7}$$

where $\sigma$ and $\mu$ represent the standard deviation and mean of the gates assigned to each expert within a batch, separately. The coefficient $\alpha$ is to balance the auxiliary objective and the main objective.

**Total Loss.** The total loss consists of the language modeling loss and auxiliary loss as follows:

$$\mathcal{L}_{total} = \mathcal{L}_{cv} + \mathcal{L}_{LM},\tag{8}$$

where $\mathcal{L}_{LM}$ is the Cross Entropy loss of language modeling in causal LMs.

## 4 EXPERIMENTS

We conduct extensive experiments to evaluate the effectiveness of our LoRA-Gen and compare it to the widely adopted LoRA-based fine-tuning method on reasoning tasks in a fair experimental setting. Furthermore, we assess the generalization capacity and system prompt compression performance of LoRA-Gen on an agent dataset, GPT4Tools (Yang et al., 2024b).

### 4.1 DATASETS AND METRICS.

**Reasoning Tasks.** We select eight widely-used benchmarks to assess the reasoning ability of LoRA-Gen across various knowledge domains ranging from natural science to daily life following counterparts (Dou et al., 2024; Li et al., 2024a). One classification task: BoolQ (Clark et al., 2019). Five question-answering tasks: ARC-c (Clark et al., 2018), ARC-e (Clark et al., 2018), OpenBookQA (Mihaylov et al., 2018), PIQA (Bisk et al., 2020) and SocialQA (Sap et al., 2019). One science completion task: Hellaswag (Zellers et al., 2019) and a fill-in-the-blank task: Winogrande (Sakaguchi et al., 2020).

**Agent Dataset.** We utilize the GPT4Tools (Yang et al., 2024b) which provides a benchmark to evaluate the ability of LLM to use tools, to access the effectiveness of LoRA-Gen in the deployment of intelligent agents. GPT4Tools constructs a tool-related instructional dataset, including positive samples, negative samples, and context samples. It consists of 71k instruction-response pairs with 21 tools in the training set and 652 items in the test set with 8 novel tools absent from the training set.

**Metrics.** The performance of all reasoning benchmarks is measured with the accuracy metric in all datasets. To further evaluate the performance in multi-task learning, we utilize two metrics: the average accuracy (AVE.) and the harmonic mean (HAR.) of all task results. For GPT4Tools, we measure the performance of the method from five aspects: successful rate of thought ($SR_t$), successful rate of action ($SR_{act}$), successful rate of arguments ($SR_{args}$), successful Rate (SR) and IoU according to GPT4Tools.

### 4.2 IMPLEMENTATION DETAILS

We deploy LLaMA3-8B (Dubey et al., 2024) as the cloud-side LM during online task-specific LoRA parameters generation. We finetune the q and v projection layers of the LLM with a LoRA adapter. The number of experts is 8 and we set K in the routing function TOP-K to 2 by default. The coefficient $\alpha$ for auxiliary loss $\mathcal{L}_{cv}$ is set 0.01. The models are trained with eight 65GB 910B NPUs in default. More details can be viewed in the Appendix.

| Method | Seen Tasks | | | | | Unseen Tasks | | | AVE. ↑ | HAR. ↑ | Latency (ms) ↓ |
|---|---|---|---|---|---|---|---|---|---|---|---|
| | ARC-c | ARC-e | OBQA | BoolQ | SIQA | HellaS | WinoG | PIQA | | | |
| TinyLlaMA-1.1B | 34.2 | 66.9 | 27.4 | 58.8 | 46.0 | **45.8** | 60.7 | 73.9 | 51.7 | 46.7 | 44.5 |
| +LoRA | 33.6 | 67.6 | 28.6 | 71.9 | 51.5 | 44.5 | 61.9 | **75.1** | 54.3 | 48.5 | 44.5 |
| +LoRAMoE | 35.2 | 68.8 | 28.6 | 73.2 | **52.1** | 45.4 | 62.0 | 74.1 | 54.9 | 49.3 | 55.9 |
| +MixLoRA | 33.5 | 67.7 | 28.4 | 73.3 | 51.4 | 44.9 | 62.3 | 74.6 | 54.5 | 48.6 | 100.1 |
| +LoRA-Gen | **35.8** | **69.1** | **30.4** | **73.6** | 49.6 | 45.5 | **62.6** | 74.1 | **55.1**$_{\pm 0.01}$ | **49.8** | **21.2** |
| Qwen-1.5B | 41.9 | 73.1 | 29.0 | 73.3 | 50.6 | 49.0 | 65.3 | 76.2 | 57.3 | 51.9 | 56.3 |
| +LoRA | 43.3 | 73.9 | 31.2 | 77.6 | **54.9** | 48.8 | 66.5 | 76.9 | 59.1 | 53.9 | 56.3 |
| +LoRAMoE | 43.9 | 73.7 | 29.8 | 77.3 | 53.4 | 48.7 | 66.3 | 76.9 | 58.8 | 53.2 | 65.7 |
| +MixLoRA | 43.4 | 73.8 | 31.8 | 78.2 | 54.6 | 48.9 | 66.4 | 76.5 | 59.2 | 54.2 | 141.9 |
| +LoRA-Gen | **44.3** | **74.3** | **33.4** | **79.6** | 53.6 | **49.1** | **67.4** | **76.9** | **59.8**$_{\pm 0.01}$ | **55.0** | **26.7** |
| Gemma-2B | 50.3 | 81.5 | 33.8 | 73.4 | 49.3 | 55.6 | 71.5 | 78.7 | 61.8 | 57.0 | 87.3 |
| +LoRA | 49.9 | 78.2 | 36.0 | **80.9** | 56.8 | 55.4 | 71.7 | 79.2 | 63.5 | 59.2 | 87.3 |
| +LoRAMoE | 50.9 | **82.0** | 38.8 | 78.4 | 55.2 | 54.0 | **72.9** | 79.3 | 63.9 | 60.0 | 101.8 |
| +MixLoRA | **52.3** | 79.4 | 38.6 | 75.6 | **59.1** | 54.1 | 72.7 | 78.2 | 63.8 | 60.2 | 177.7 |
| +LoRA-Gen | 51.2 | 81.9 | **39.0** | 76.2 | 55.6 | **56.0** | 71.6 | **79.5** | **63.9**$_{\pm 0.01}$ | 60.2 | **36.1** |

Table 2: Comparison of the performance with 5-shot samples on various reasoning benchmarks. Seen tasks indicate that the datasets are part of the training set, while unseen tasks are not. AVE denotes the average accuracy of 8 tasks while HAR is the harmonic mean. The latency scores of various methods are all calculated on ARC-c. Latency is measured on a Nvidia A100 GPU.

| Method | W/ Training | W/ Tools Definiton | $SR_t$ | $SR_{act}$ | $SR_{args}$ | SR | IoU | Average Score ↑ | Compress Ratio ↑ |
|---|---|---|---|---|---|---|---|---|---|
| Gemma-2B | ✗ | ✔ | 86.3 | 77.6 | 77.7 | 65.0 | 89.7 | 79.3 | 1x |
| +LoRA | ✔ | ✔ | **99.4** | 79.6 | **93.8** | 78.2 | 91.0 | 88.4 | |
| +LoRA | ✔ | ✗ | 98.0 | 60.9 | 83.2 | 52.1 | 81.3 | 75.1 | |
| +LoRA-Gen | ✗ | ✗ | 94.1 | 86.8 | 79.7 | 73.3 | 86.9 | 84.2 | 10.1x |
| +LoRA-Gen | ✔ | ✗ | 98.6 | **88.0** | 93.4 | **84.0** | **93.6** | **91.5** | |

Table 3: Performance of different fine-tuning strategies with Gemma-2B (Team et al., 2024) on test set of GPT4Tools (Yang et al., 2024b). W/ Training denotes Gemma-2B is fine-tuning on the training set of GPT4Tools with vanilla LoRA or our LoRA-Gen. Gray rows indicate scenarios where the system prompt does not contain tools definitions, typically constituting 91% of the input context.

## 4.3 MAIN RESULTS

**Reasoning Tasks.** We first evaluate the performance of LoRA-Gen in the reasoning scenario as shown in Table 2. We divide eight commonly used datasets into two parts, one as the multi-task learning set, including ARC-c, ARC-e, OpenBookQA, BoolQ, SocialQA and the other as an unseen test set, including Hellaswag, Winogrande and PIQA. We randomly sample to construct multi-shot training data. As shown in Table 2, LoRA-Gen consistently achieves comparable performance while exhibiting lower latency compared to other fine-tuning methods across various backbone models. Specifically, LoRA-Gen with TinyLlaMA (Zhang et al., 2024a) achieves 55.1 AVE and 49.8 HAR, surpassing LoRA and MixLoRA. Similar trends are seen in Qwen-1.5B (Yang et al., 2024a) and Gemma-2B (Team et al., 2024). Notably, in the case of 5-shot inference, our paradigm is theoretically capable of achieving competitive performance with just one-sixth of the sequence length. In practice, LoRA-Gen significantly reduces the latency from 44.5 ms to 21.2 ms for TinyLlaMA and from 87.3 to 36.1 ms for Gemma-2B, boosting other methods. The results underscore the advantage of using LoRA-Gen, which balances effectiveness and efficiency across both seen and unseen tasks.

**Intelligent Agent Scenario.** We evaluate the performance of LoRA-Gen with edge-side model Gemma-2B on the GPT4Tools benchmark (Yang et al., 2024b). The results in Table 3 present a comparison of successful rates, intersection-over-union (IoU), average performance, and compress ratio (speedup). One key advantage of LoRA-Gen is to compress the tools definition within the system prompt into the generated LoRA parameters via a single-turn inference. It significantly re-

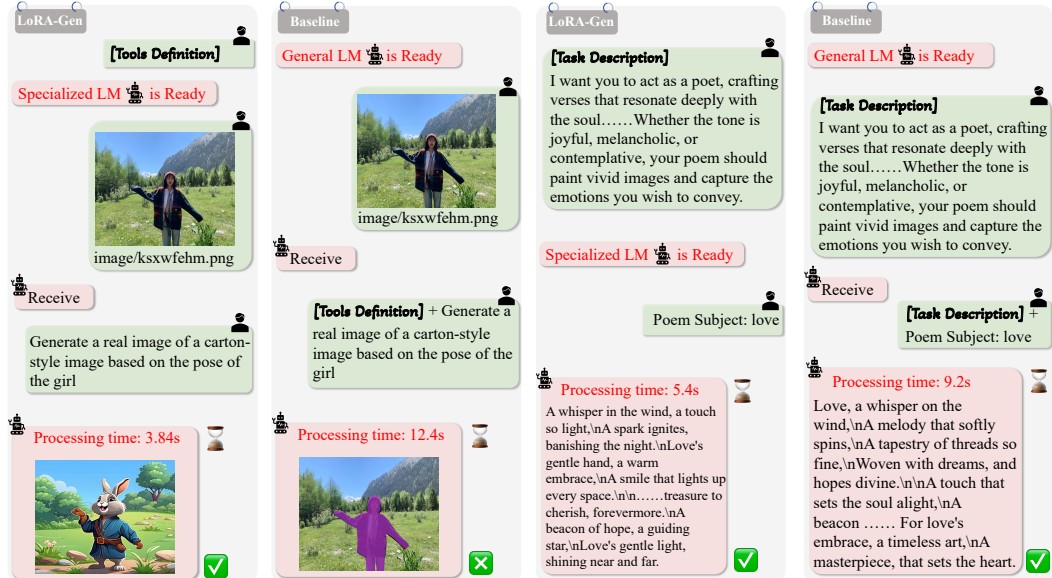

Figure 4: Visualization comparison between LoRA-Gen and baseline, Gemma-2B (Team et al., 2024). LoRA-Gen compresses the tools definition and task description into the generated LoRA parameters, effectively specializing the language model to reduce processing times while maintaining comparable performance. The detailed LM outputs and system prompt can be accessed in the Appendix.

duces the context length with a compress ratio of 10.1x, which maintains comparable performance of 91.5% average score. On the other hand, our method without training on GPT4Tools boosts original Gemma-2B by 4.9% in average score, which shows the effective generalization of our method. In contrast, removing the tool definitions in the vanilla LoRA setting leads to a marked reduction in performance (SR: -26.1%, IoU: -9.7%). Furthermore, benefiting from knowledge injection from the cloud-side language model, it surpasses the baseline by 3.1 points while maintaining a 10.1x compression ratio. The results highlight the strengths of LoRA-Gen in effectiveness and efficiency, attributed to its inference-time specializing and generalization ability to unseen tools, making it well-suited for tasks with extensive prefix descriptions.

## 4.4 ABLATION STUDY

We conduct a series of experiments to ablate the components of LoRA-Gen with Qwen-1.5B (Yang et al., 2024a) and evaluate with 5-shot samples by default. For all experiments of this section, we select ARC-C, ARC-E, OpenbookQA, and SocialQA as the seen tasks, while Winogrande and PIQA represent the unseen tasks.

**Number of Experts in Online Expert Pool.** As shown in Table 4, we present the performance of different numbers of experts in the cloud-side LoRA pool. Performance generally improves with an increasing number of experts. With 4 experts, the AVE. is 56.1%, and the HAR. is 49.9 %. Increasing the number of experts to 12 yields slight improvements, with the AVE. rising to 57.0% and the HAR. to 50.7%. However, the best performance is achieved with 8 experts, where both AVE. (58.7%) and HAR. (53.6%) reach their peak values. This may indicate that 8 experts strike the best balance between multi-task learning and unseen generalization.

**Exploration of Balanced Load Strategy.** Ensuring a balanced load of experts can significantly improve the robustness and stability of the model. We initially conduct an ablation study to assess the impact of the absence of auxiliary losses on model performance. Without the auxiliary loss, the AVE. decreases by 2.6 points. Subsequently, we summarize the impact of different values of coefficient for auxiliary loss as shown in Table 5. As the auxiliary loss coefficient decreases, a sig-

| Expert Number | AVE. | HAR. |
|---|---|---|
| 4 | 56.1 | 49.9 |
| 12 | 57.0 | 50.7 |
| 8 | **58.7** | **53.6** |

Table 4: **Number of Experts in cloud-side pool.**

| Auxiliary Loss Coefficient | AVE. | HAR. |
|---|---|---|
| 0.1 | 57.1 | 51.7 |
| 0.005 | 56.8 | 50.5 |
| 0.01 | **58.7** | **53.6** |

Table 5: **Different coefficient impact of auxiliary loss.**

| Few-shot Number | AVE. | HAR. |
|---|---|---|
| 3-shot† | 55.5 | 49.3 |
| 5-shot† | 56.0 | 49.9 |
| 1-shot‡ | **58.4** | **53.4** |

Table 6: † **is baseline and** ‡ **denotes using LoRA-Gen.**

| Router Strategy | AVE. | HAR. |
|---|---|---|
| GumbleTOP-K | 56.4 | 51.2 |
| KeepTOP-K | **58.7** | **53.6** |

Table 7: **Different router strategy for online experts in the cloud-side LoRA pool.**

| LoRA Generation | Seen AVE. | Unseen AVE. |
|---|---|---|
| Direct | **52.4** | 61.0 |
| Indirect | 52.0 | **72.1** |

Table 8: **Effectiveness of indirect LoRA generation via meta-token.**

nificant improvement in both performance metrics is observed. Reducing the coefficient from 0.1 to 0.01 yields further gains, resulting in an average (AVE) of 58.7% and a harmonic mean (HAR) of 53.6%, thereby achieving an optimal balance between the auxiliary strategy and the primary objective function. In addition to incorporating auxiliary training objectives, we also investigate whether the router function based on the Gumbel distribution can achieve a more balanced load of various experts without a performance drop. As illustrated in Table 7, we compare two routing strategies employed for online experts within the cloud-side LoRA pool. The GumbelTOP-K strategy results in an average (AVE) of 56.4% and a harmonic mean (HAR) of 51.2%. In contrast, the KeepTOP-K strategy exhibits a notable enhancement, attaining an AVE of 58.7% and a HAR of 53.6%. We consider that an overabundance of randomness may impair the capacity of experts to learn specific tasks during the optimization process.

**Effectiveness of Meta Token.** We attempt to utilize the cloud-side large language model to generate LoRA parameters in a single forward pass directly instead of meta tokens. Specifically, we directly transform the output tokens of LLM to the LoRA weights space with a feedforward neural network and get the $i$-th layer generated LoRA weights $\in \mathbb{R}^{3 \times 2 \times d \times r}$, where $d$ is the hidden dimension and $r$ denotes the low rank of LoRA. As indicated by the experimental results in Table 8, this approach exhibits comparable performance to that achieved through meta tokens on the seen tasks, while the results on the unseen tasks are significantly lower than those obtained with meta tokens, trailing by 11.1%. Generating LoRA parameters directly leads to pronounced overfitting to the training domain, caused by the large parameter space, thereby limiting its ability to generalize to unseen tasks.

**Effectiveness of Knowledge Transfer.** As depicted in Table 6, we compare the performance of the baseline model and our LoRA-Gen across different few-shot samples. Remarkably, LoRA-Gen with just a 1-shot sample surpasses the baseline with 5-shot samples by 3.5% on HAR. We attribute this to the use of LLaMA3-8B (Dubey et al., 2024) as the cloud model, which transfers a portion of its knowledge to the edge-side language model via reparameterization.

## 4.5 QUALITATIVE STUDY IN AGENT SCENARIO

We deploy LoRA-Gen within Gemma-2B and conduct case studies and visualizations. As illustrated in Figure 4, LoRA-Gen removes the 26 tools description from the input of the model, significantly reducing inference time and achieving a 3.2x speedup compared to the baseline. The limited generalization of the baseline model results in incorrect tool selection, thereby highlighting the effectiveness of our method. Additionally, in the open text generation scenario, LoRA-Gen accelerates reasoning time by compressing the task definition while achieving comparable results. The corresponding generation results are detailed in the appendix.

## 5 CONCLUSION

In this paper, we propose an online LoRA generation framework, called LoRA-Gen, which utilizes a cloud-side language model to generate task-specific LoRA parameters for edge-side models. Our strategy offers four advantages over previous methods, including context compression for unseen tasks, a reparameterized language model, inference-time specializing, and knowledge transfer. Extensive experiments show that LoRA-Gen achieves competitive results and an impressive speedup on common-sense reasoning tasks. Additionally, our method achieves a compress ratio of 10.1x on zero-shot agent tasks, indicating its potential applicability to more scenarios.

**Limitations.** Our method supports diverse application scenarios, such as tool calls, personalized virtual assistants, offline intelligent systems, IoT device control, and tasks necessitating long system prompts. However, the current paradigm needs to predefine a pair of cloud and edge-side LM. The model-agnostic framework leaves an open question for future work.

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

## A  APPENDIX

### A.1  TRAINING DETAILS

| Hyper-parameters | LoRA-Gen |
|---|---|
| optimizer | AdamW |
| learning rate | 2e-5 |
| warm steps | 50 |
| weight decay | 0.1 |
| optimizer momentum | $\beta_1, \beta_2$=0.9, 0.999 |
| batch size | 64 |
| epoch | 4 |
| max length | 2048 |
| LoRA (Hu et al., 2021) attention dimension (r) | 16 |
| LoRA (Hu et al., 2021) scaling alpha ($\alpha$) | 16 |
| LoRA (Hu et al., 2021) drop out | 0.05 |

Table 9: Fine-tuning configuration.

The models are trained with eight NPUs (64GB memory per device) by default. We set betas and momentum of the AdamW optimizer with (0.9, 0.999) and 0.9, respectively. During training, we utilize a Cosine Scheduler with an initial learning rate of $2 \times 10^{-5}$ and weight decay of 0.1. The details are shown in Table 9

### A.2  DETAILED ASSISTANT OUTPUT

Tools definition is shown in Table 15, Table 16 and Table 17. Task description for the role play in qualitative study of main text can be seen in Table 14. To strengthen LoRA-Gen's ability to compress and process instructions in system prompt, we modify the Alpaca dataset, using GPT-4 to generalize specific problems into instruction sets, which are subsequently used as training data.

### A.3  STATISTICAL SIGNIFICANCE

| Method | ARC-c | ARC-e | OBQA | BoolQ | SIQA | HellaS | WinoG | PIQA |
|---|---|---|---|---|---|---|---|---|
| TinyLLaMA | 0.0146 | 0.0089 | 0.0219 | 0.0076 | 0.0112 | 0.0050 | 0.0134 | 0.0100 |
| Qwen | 0.0145 | 0.0089 | 0.0229 | 0.0071 | 0.0113 | 0.0050 | 0.0132 | 0.0098 |
| Gemma | 0.0146 | 0.0089 | 0.0218 | 0.0075 | 0.0112 | 0.0050 | 0.0135 | 0.0096 |

Table 10: **Standard error on language model benchmarks.**.

The standard errors of different tasks are shown in Table 10, all statistics are calculated with the open-sourced lm-evaluation-harness project (Gao et al., 2024b). Additionally, we have re-evaluated our method 4 times on GPT4Tools with a variation of about 0.65% in average score.

## A.4 TRAINING DATA.

Table 11 outlines the data scale for each reasoning tasks. Moreover, we process the Alpaca dataset through GPT-4, resulting in a filtered and abstracted set of 37,658 training samples.

| Method | ARC-c | ARC-e | OBQA | BoolQ | SIQA | HellaS | WinoG | PIQA |
|---|---|---|---|---|---|---|---|---|
| Train | 1120 | 2250 | 4957 | 9427 | 33410 | 39905 | 9248 | 16100 |
| Test | 1171 | 2380 | 500 | 3270 | 1954 | 10042 | 1267 | 1838 |

Table 11: **The data size of tasks used in our experiments.**

## A.5 EFFICIENCY COMPARISON

| Method | Training Mode | | | Inference Mode | | |
|---|---|---|---|---|---|---|
| | FLOPs | Memory | Latency | FLOPs | Memory | Latency |
| +LoRA | 4.736E+11 | 37096MiB | 0.85s | 4.708E+11 | 11208MiB | 0.19s |
| +LoRAMoE | 4.742E+11 | 26326MiB | 1.19s | 4.742E+11 | 11286MiB | 0.22s |
| +MixLoRA$^{\dagger}$ | 5.061E+11 | 30844MiB | 2.17s | 5.048E+11 | 11828MiB | 1.08s |
| +LoRA-Gen | 1.667E+12 | 39603MiB | 2.84s | 1.552E+11 | 10932MiB | 0.11s |

Table 12: **Efficiency Comparison.**

Table 12 presents the efficiency Comparison among different approaches. MixLoRA$^{\dagger}$ indicates the method without specific optimization. All metrics are measured on a Nvidia A100 GPU. FLOPs are measured using an input of 100 tokens and an instruction of 200 tokens, while memory and latency are evaluated in training mode with a batch size of 8 per GPU.

## A.6 COMPARISON WITH AUTOCOMPRESSORS

| Method | HellaS | WinoG | PIQA | Ave. | Latency |
|---|---|---|---|---|---|
| AutoCompressors (Chevalier et al., 2023) | 44.7 | 62.4 | 73.3 | 60.1 | 11.4ms |
| LoRA-Gen | 46.3 | 63.7 | 74.9 | 61.6 | 7.54ms |

Table 13: **Comparison with AutoCompressors based on OPT-2.7B..**

As shown in Table 13, our method achieves a 1.4-point improvement in average accuracy of unseen tasks over AutoCompressors (Chevalier et al., 2023). Additionally, the bootstrap test interval of HellaS is [0.0022, 0.0297], which confirms a statistically significant enhancement compared to AutoCompressors. Furthermore, we measure the latency of AutoCompressors, with the results showing a 1.52x speedup that emphasizes the efficiency of our approach.

**Task description of role play:**

I want you to act as a poet, crafting verses that resonate deeply with the soul. Your task is to create a poem that not only evokes emotions but also possesses the power to move and inspire those who read it. The subject matter will be provided by the user, and your words should be carefully chosen to express the essence of that theme in a way that touches hearts and stirs deep feelings. Aim for beauty in your language—let your choice of words carry both meaning and emotion, so that each line speaks to the reader on a profound level. Whether the tone is joyful, melancholic, or contemplative, your poem should paint vivid images and capture the emotions you wish to convey.

**Edge-side model with LoRA-Gen answer:**

A whisper in the wind, a touch so light,\A spark ignites, banishing the night.\Love's gentle hand, a warm embrace,\A smile that lights up every space.\\Two souls entwined, a tapestry of dreams,\Where laughter dances and love softly gleams.\A bond unbreakable, a trust so deep,\In each other's eyes, secrets they keep.\\Through storms and sunshine, hand in hand they roam,\Building castles of hope, a love that calls them home.\A silent language, spoken in the heart,\A symphony of feelings, a work of art.\\For love is a journey, a path to explore,\A treasure to cherish, forevermore.\A beacon of hope, a guiding star,\Love's gentle light, shining near and far.

**The baseline answer:**

Love, a whisper on the wind,\A melody that softly spins,\A tapestry of threads so fine,\Woven with dreams, and hopes divine.\\A touch that sets the soul alight,\A beacon in the darkest night,\A warmth that melts the coldest fear,\A solace, ever drawing near.\\It blooms in laughter, shared and bright,\In silent moments, bathed in light,\A gentle hand, a knowing glance,\A shared embrace, a sacred dance.\\Love's strength endures, a steadfast guide,\Through storms and calm, it cannot hide,\A constant echo, soft and deep,\A promise whispered, secrets to keep.\\So let us cherish every breath,\This precious gift, beyond all death,\For love's embrace, a timeless art,\A masterpiece, that sets the heart. \\\Let me know if you'd like me to explore a specific aspect of love, or if you have a particular style or tone in mind.

Table 14: Detailed supplement to the visualization results in the main text.

**Tools definition 1**

GPT4Tools can handle various text and visual tasks, such as answering questions and providing in-depth explanations and discussions. It generates human-like text and uses tools to indirectly understand images. When referring to images, GPT4Tools follows strict file name rules. To complete visual tasks, GPT4Tools uses tools and stays loyal to observation outputs. Users can provide new images to GPT4Tools with a description, but tools must be used for subsequent tasks.\nTOOLS:\n——\n\nGPT4Tools has access to the following tools:\n\n> Assess the Image Quality: useful when you want to give a quality score for the input image. like: assess a quality score for this image, what is the quality score of this image, or can you give a quality for this image. The input to this tool should be a string, representing the image_path.\n> Recognize Face: Useful when you only want to recognize faces in the picture. like: recognize who appears in the photo. The input to this tool should be a string, representing the image_path.\n> Detect Face: Useful when you only want to detect out or tag faces in the picture. like: find all the faces that appear in the picture. tag someone in the picture. The input to this tool should be a string, representing the image_path.\n> Crop the Given Object: Useful when you want to crop given objects in the picture. The input to this tool should be a comma separated string of two, representing the image_path, the text description of the object to be cropped.\n> Generate 3D Asset From User Input Text: Useful when you want to generate an 3D assert from a user input text and save it to a file. like: generate a 3D assert of an object or something. The input to this tool should be a string, representing the text used to generate the 3D assert.\n> Image Super-Resolution: Useful when you want to enhance the resolution and quality of low-resolution images. like: enhance this image, restore this image. The input to this tool should be a string, representing the image_path.\n> Detection: Useful when you want to detect all objects of the image, but not detect a certain object according to the text. like: detect all the objects in this image, or detect this image. The input to this tool should be a string, representing the image_path.\n> Text Detection On Image: Useful when you want to detect the text in the image. The input to this tool should be a string, representing the image_path.\n> Generate Image From User Input Text: useful when you want to generate an image from a user input text and save it to a file. like: generate an image of an object or something, or generate an image that includes some objects. The input to this tool should be a string, representing the text used to generate image.\n> Generate Image Condition On Canny Image: useful when you want to generate a new real image from both the user description and a canny image. like: generate a real image of a object or something from this canny image, or generate a new real image of a object or something from this edge image. The input to this tool should be a comma separated string of two, representing the image_path and the user description.\n> Generate Image Condition On Depth: useful when you want to generate a new real image from both the user description and depth image. like: generate a real image of a object or something from this depth image, or generate a new real image of a object or something from the depth map. The input to this tool should be a comma separated string of two, representing the image_path and the user description\n> Segment the Image: useful when you want to segment all the part of the image, but not segment a certain object.like: segment all the object in this image, or generate segmentations on this image, or segment the image,or perform segmentation on this image, or segment all the object in this image.The input to this tool should be a string, representing the image_path\n> Generate Image Condition On Sketch Image: useful when you want to generate a new real image from both the user description and a scribble image or a sketch image. The input to this tool should be a comma separated string of two, representing the image_path and the user description.\n> Replace Something From The Photo: useful when you want to replace an object from the object description or location with another object from its description. The input to this tool should be a comma separated string of three, representing the image_path, the object to be replaced, the object to be replaced with\n>

Table 15: Detailed supplement to the visualization results in the main text.

**Tools definition 2**

Generate Image Condition On Segmentations: useful when you want to generate a new real image from both the user description and segmentations. like: generate a real image of a object or something from this segmentation image, or generate a new real image of a object or something from these segmentations. The input to this tool should be a comma separated string of two, representing the image_path and the user description$\backslash n>$ Generate Image Condition On Pose Image: useful when you want to generate a new real image from both the user description and a human pose image. like: generate a real image of a human from this human pose image, or generate a new real image of a human from this pose. The input to this tool should be a comma separated string of two, representing the image_path and the user description$\backslash n>$ Instruct Image Using Text: useful when you want to the style of the image to be like the text. like: make it look like a painting. or make it like a robot. The input to this tool should be a comma separated string of two, representing the image_path and the text.$\backslash n>$ Generate Image Condition On Soft Hed Boundary Image: useful when you want to generate a new real image from both the user description and a soft hed boundary image. like: generate a real image of a object or something from this soft hed boundary image, or generate a new real image of a object or something from this hed boundary. The input to this tool should be a comma separated string of two, representing the image_path and the user description$\backslash n>$ Generate Image Condition On Normal Map: useful when you want to generate a new real image from both the user description and normal map. like: generate a real image of a object or something from this normal map, or generate a new real image of a object or something from the normal map. The input to this tool should be a comma separated string of two, representing the image_path and the user description.$\backslash n>$ Remove Something From The Photo: useful when you want to remove and object or something from the photo from its description or location. The input to this tool should be a comma separated string of two, representing the image_path and the object need to be removed.$\backslash n>$

Table 16: Detailed supplement to the visualization results in the main text.

**Tools definition 3**

Generate Image Condition On Normal Map: useful when you want to generate a new real image from both the user description and normal map. like: generate a real image of a object or something from this normal map, or generate a new real image of a object or something from the normal map. The input to this tool should be a comma separated string of two, representing the image_path and the user description$\backslash n>$ Segment the Image: useful when you want to segment all the part of the image, but not segment a certain object.like: segment all the object in this image, or generate segmentations on this image, or segment the image,or perform segmentation on this image, or segment all the object in this image.The input to this tool should be a string, representing the image_path$\backslash n>$ Get Photo Description: useful when you want to know what is inside the photo. receives image_path as input. The input to this tool should be a string, representing the image_path.$\backslash n>$ Edge Detection On Image: useful when you want to detect the edge of the image. like: detect the edges of this image, or canny detection on image, or perform edge detection on this image, or detect the canny image of this image. The input to this tool should be a string, representing the image_path$\backslash n>$ Predict Depth On Image: useful when you want to detect depth of the image. like: generate the depth from this image, or detect the depth map on this image, or predict the depth for this image. The input to this tool should be a string, representing the image_path$\backslash n>$ Replace Something From The Photo: useful when you want to replace an object from the object description or location with another object from its description. The input to this tool should be a comma separated string of three, representing the image_path, the object to be replaced, the object to be replaced with$\backslash n\backslash n$

Table 17: Detailed supplement to the visualization results in the main text.

