# OpenReview forum: "LoRA-Gen: Specializing Language Model via Online LoRA Generation"
_ICLR.cc/2025/Conference — ICLR 2025 Conference Withdrawn Submission_

### Official Review · Reviewer_WeXV · 2024-11-02

**Soundness:** 2
**Presentation:** 3
**Contribution:** 3
**Rating:** 5
**Confidence:** 4

**Summary:**

This work proposes LoRA-Gen, a method for generating LoRA modules for reduced scale LMs using LLMs running in the cloud. Given a prompt, the method generates tokens using the LLM based on the system prompt which the method is aiming to encode. Then, the method uses these so-called "meta-tokens" to figure out weights over a set of pre-trained LoRA modules from a set of seen tasks. These LoRA weights are then distributed to the small LLM, merged in to minimize latency impact, and then used for task inference on device.

**Strengths:**

- The core problem statement of "utilizing a large cloud-side model to generate parameters for a smaller edge-side model to achieve
better specialization" is a very reasonable motivator for on-device specialization with fewer shortcomings from smaller models. It's also a realistic setting that could exist for real-world deployments. I haven't seen this formulation in related work and it seems a strong setting of interest.

- Hyper-networks for LoRA modules are a very interesting subject area, as they highlight, since they can enable the projection to unseen tasks which is evaluated here. I think this premise of work also has strong advantages and seems worth exploring.

**Weaknesses:**

- The work is missing ablations/comparisons to the other context compression methods which they list in section 2.3, an area which was introduced for LLMs even earlier than they reference in their related works by Snell et al. in 2022. Given that most of the latency gains come primarily from using context compression, it would be necessary to see that the compression ratio and latency of *at least one other* context compression method is worse than the proposed approach to assess whether these gains are substantial v.s. prior work.

- The work doesn't engage with a clear weakness of this method which is that it requires more compute at training time than any of the baselines due to the use of the larger cloud-side model. I'd like to see a concrete comparison of the training compute cost between these different methods included so that readers can understand the degree of training compute increase.

- The work does not perform any form of statistical significance testing on the results nor does it report how data was held out for hyperparameter selection. Both of these together present a significant soundness concerns if they are not addressed.

Each of these items has associated questions below.

**Questions:**

Questions:

- In Section 2.3, you cover many context distillation methods such as gisting, AutoCompressors, ICAE, and 500xCompressor. Why were these not included as baselines? Especially depending on the LoRA hyperparameters, the compression ratios of such methods may be perhaps even better than LoRA-Gen so the omission of these baselines despite your awareness of them requires some justification or, preferably, at least one of these methods should be compared to. This is especially true since at least some of these methods, such as gisting, are conceptually much simpler than the method proposed here.

- How many FLOPs does it take to train LoRA Gen? How many FLOPs does it take to train each of the other baselines? In addition to inference latency, these training FLOPs are a key attribute of each method that isn't currently reported.

- Using a bootstrap significance test, is LoRA Gen a significant improvement over the baselines reported in Table 2? You can find methodological best practices for significance testing here: https://aclanthology.org/P18-1128/

- What are the validation set results for all the experiments in section 4.4? These *must* be included in the appendix to show that the hyperparameters used in the paper would be selected using validation set results rather than the test set results currently listed.

- On line 100, you reference "additionally, since it does not require the input of agent definitions during inference, it achieves a remarkable 10.1x speedup". Is this referring to the 10.1x compression ratio on the prompt? If so, this statement seems inaccurate since the prompt tokens have a lower impact on latency than generated tokens due to batching. This is doubly shown by your own latency numbers in Table 2, where LoRA-Gen reduces the latency only by 2.4x which is notably far less than 10.1x.

- I'm still a bit unclear on the term "meta-tokens" from the cloud model. As far as I can tell, these tokens are not specialized in any way for this parameter generation task but are just regular llama tokens given the prompt. Is this correct? If so, it should be explained as such rather than introducing more terminology needlessly. If they have more customized use than this, especially to encourage them to have mappings to specific layers in the downstream model, this requires a lot more explanation than the description on 216-220.

- The caption and description of Figure 1 is a bit unclear to me in the current draft. If I'm understanding correctly, the few-shot examples listed here are being used as a system prompt which is why even the baseline method is slower than LoRA-Gen. Is this correct? If so, you should make this clearer by listing that item as "Prompted Qwen"! Furthermore, the prompted method is slower at inference, but the inference cost is the *only* cost, while LoRA-Gen has a large training cost right? Do you think it's fair to compare these metrics solely along this axis, without marking that some of the methods are training free?


Misc. Typos and Suggestions:
- Throughout the work, the `\citep` tag should be used much more frequently. As a rule of thumb, if you aren't using the name of the authors as part of the text that fits grammatically, the citation should be included in parentheses. For example, on lines 148 and 149 all of these citations should be with `citep` rather than `cite`.
-  LLaMA3 Touvron et al. (2023b). This is the incorrect citation for Llama 3. The work cited is the original Llama paper, LLama 3 is Dubey et al. 2024 https://arxiv.org/abs/2407.21783
- Line 218 "by generates" should be "by generating".

---

> ### Author Response · Authors · 2024-11-23
> **Rebuttal by authors 1/2**
>
> **Q1: Missing ablations/comparisons to the other context compression methods.**
>
> **Ans:**
> We focus on mitigating the efficiency constraints observed in previous LoRA-MoE-based methods and provide a new perspective that a large cloud-side model generates parameters for a smaller edge-side model, enabling improved specialization. While this approach shares similarities with compression methods, our primary focus is on enhancing generalization for unseen tasks through meta-learning strategies, alongside improving operational efficiency. As shown in Table 20, our comparisons with the methods mentioned by the reviewer demonstrate that our method achieves superior performance while maintaining competitive latency.
>
> | Base Model      | Method               | HellaS | WinoG | PIQA  | Average | Latency   | Latency w/o instruction |
> |------------------|----------------------|:--------:|:-------:|:-------:|:---------:|:-----------:|:------------------:|
> | LLaMA-7B        | +Gisting             | 19.6   | 38.6  | 46.1  | 34.8    | 166.3ms   | 161.8ms          |
> |                  | +LoRA-Gen           | 58.1   | 72.1  | 77.2  | 69.1    | 162.0ms   |                  |
> | LLaMA2-7B       | +AutoCompress        | 57.3   | 68.8  | 77.5  | 68.9    | 64.08ms   | 61.32ms          |
> |                  | +500xCompress†      | 25.9   | 48.1  | 52.7  | 42.3    | 76.87ms   |                  |
> |                  | +ICAE†              | 26.7   | 48.6  | 55.9  | 43.7    | 69.76ms   |                  |
> |                  | +LoRA-Gen           | 56.1   | 72.5  | 78.4  | 69.0    | 61.49ms   |                  |
>
> **Table 20: Performance Comparison on unseen tasks with 5-shot samples among compression methods. † indicates that the method is without pretraining due to their weights not being publicly available. The latency is measured on an Nvidia V100 GPU.**
>
>
> **Q2: Comparison of more efficiency metrics.**
>
> **Ans:**
> Please refer to the response for Reviewer PsM9's Q1, and we have already presented it in the A.5 of our Appendix.
>
> **Q3: Statistical significance testing.**
>
> **Ans:**
> Thanks for the great advice. The standard error is shown in A.3 of Appendix.
> The results of the bootstrap significance test are shown in Table below.
> Our approach seeks to explore a novel perspective on LoRA MoE, focusing on enhancing generalization capabilities and reasoning efficiency while maintaining comparable performance.
>
> | Confidence Interval   | LoRA    | LoRAMoE | MixLoRA | Gisting | AutoCompress | 500xCompress | ICAE   |
> |------------------------|:---------:|:---------:|:---------:|:---------:|:--------------:|:--------------:|:--------:|
> | Lower Value   | -0.1395 | -0.1375 | -0.1401 | 0.1967  | -0.1253      | 0.1090       | 0.0860 |
> | Higher Value   | 0.1695| 0.1751 | 0.1688 | 0.5007  | 0.1580     | 0.4363       | 0.4293 |
>
> **Table 21:The lower value of confidence interval in bootstrap testing.**
>
>
> **Q4: Validation set results for all the experiments in section 4.4.**
>
> **Ans:**
> We follow the counterparts (MixLoRA and LoRAMoE) to construct the evaluation settings that utilize the same set of datasets.
> The official datasets SIQA, WinoG, and PIQA do not contain valid parts, so we utilize the results of the test set to calculate the average score.
> Additionally, we take the ablation experiments of the Auxiliary Loss Coefficient as an example.
> The results on both the validation and test sets, as shown in Table 22, exhibit a consistent performance trend.
>
> | Set    | Loss Coefficient | ARC-c | ARC-e | OBQA  | SIQA$^*$ | WinoG$^*$ | PIQA$^*$ | AVE. $\uparrow$ | HAR. $\uparrow$ |
> |--------|:------------------:|:-------:|:-------:|:-------:|:----------:|:-----------:|:----------:|:-----------------:|:-----------------:|
> | Test   | 0.005            | 41.6  | 72.8  | 28.8  | 54.5     | 66.9      | 76.3     | 56.8            | 50.5            |
> |        | w/o              | 39.8  | 71.5  | 29.0  | 53.0     | 66.2      | 76.8     | 56.1            | 49.8            |
> |        | 0.01             | 44.8  | 74.7  | 33.0  | 55.3     | 67.3      | 76.8     | 58.7            | 53.6            |
> | Val    | 0.005            | 42.3  | 72.7  | 28.8  | 54.5     | 66.9      | 76.3     | 56.9            | 50.6            |
> |        | w/o              | 39.5  | 71.3  | 28.6  | 53.0     | 66.2      | 76.8     | 55.9            | 49.5            |
> |        | 0.01             | 44.5  | 74.7  | 33.6  | 55.3     | 67.3      | 76.8     | 58.7            | 53.7            |
>
> **Table 22: Validation set results.**
>
> **Q5: Is this referring to the 10.1x compression ratio on the prompt?**
>
> **Ans:** The 10.1x metric indeed reflects our token compression rate. As the <EOS> character in the generation mode differs across models and is not manually standardized, compression rate serves as a consistent evaluation metric.
> We have modified the manuscript in line 101.

---

> > ### Comment · Reviewer_WeXV · 2024-11-25
> > **On Statistical Testing**
> >
> > Thanks for you efforts to add statistical significance testing! I'm a bit confused as to the contents of Table 21, as it does not specify which of your tasks this is with respect to! Is it one of the tasks in particular or is it one of the aggregated metrics?
> >
> > The concerning factor here seems to be that for LoRA, LoRAMoE, MixLoRA, and AutoCompress the confidence interval covers 0 and the null-hypothesis cannot be excluded. This makes it especially important to understand which metric this table is from.
> >
> > As for the standard errors in A.3, thank you for adding these! However, to interpret these it would be necessary to specify which model they correspond to in Table 2. Perhaps more helpful even than that would be to include these directly in Table 2 with a (+-).

---

> > ### Comment · Reviewer_WeXV · 2024-11-25
> > **On Prompt Compression Baselines**
> >
> > Thank you for incorporating these baselines! I'm not sure I share the conclusion that Table 20 illustrates that your "method achieves superior performance". The performance between your method and AutoCompress seems notably similar  (within the margin of error) as do the latency metrics.
> >
> > Given the closeness of these results on only the Unseen task scenario, I think full results with AutoCompress are a necessary inclusion in Figure 1 and in Table 2.

---

> > ### Comment · Reviewer_WeXV · 2024-11-25
> > **Thanks and Overall Update After Response**
> >
> > Thank you for your response! I appreciate the effort on the additional experiments. My questions on the possible test set tuning are addressed by the validation set results.
> >
> > However, overall my main concern points in the weaknesses remain open in concerning ways:
> > - The added statistical testing seems to hint that many of the improvements are indeed not statistically significant.
> > - The added baselines show that there are existing methods that seem to perform competitively to the proposed method on **both** performance and latency.
> > - The added discussion of Figure 1 opens further questions of whether the latency comparison is accurate as is, since it is unclear how the inference cost of the larger model is being incorporated into the latency metrics for the figure.

---

> > > ### Author Response · Authors · 2024-11-26
> > > **Further response to Reviewer WeXV by authors**
> > >
> > > **Q1: Questions on "Meta-Tokens"**
> > >
> > > **Ans:**
> > > We apologize for the confusion. Here is our more detailed explanation:
> > > We first construct a special token, termed <$meta$>, which is added to the tokenizer of the cloud-side LM, and the LM is finetuned with lightweight LoRA adapters (refer to sec4.2). Then we append $L$ special tokens behind the instruction to the cloud-side LLM. We obtain the final hidden state of these meta tokens as the representation in a single forward pass without decoding. We have corrected this statement instead of "autoregressive manner" in the manuscript. Furthermore, a routing module is employed to process the representation for the following weights generation refer to Figure 2.
> > >
> > > **Q2: Questions on Statistical Testing.**
> > >
> > > **Ans:**
> > > 1. Bootstrap testing only consists of the accuracy results of all tasks (Eight tasks for the  LoRA-based method in Table 2 and three tasks for the compression method in Table 20) with 10,000 sample times.
> > > We aim to emphasize our main contribution relies on introducing a novel approach to enhance the efficiency of LoRA-MoE-based strategies (5.3x speedup than MixLoRA and 2.5x speedup than LoRAMoE with Qwen as shown in Table 2) while achieving comparable performance.
> > > 2. The current standard deviation corresponds to TinyLLaMA. Additionally, we have updated the results for Qwen and Gemma in the appendix and modified the average std in Table 2 with (+/-).
> > >
> > > **Q3: Questions on prompt compression baselines.**
> > >
> > > **Ans:**
> > > Our method exhibits a clear performance advantage over other compression methods(such as Gisting, ICAE, and 500xCompress).
> > > Moreover, our advantage over AutoCompress also consists of the edge-side device (CPU) speed and token compression ratio as shown in Table 23.
> > > We calculate the token numbers from the average token count of the Hella, Wing, and PIQA datasets (used in Table 20). Latency calculations follow the procedure used in Table 20 but are performed on the CPU. Nevertheless, we would like to underscore the improvements of our method in the LoRA-MoE-based approaches, highlighted in Table 1 of the manuscript, which are our core contributions.
> > >
> > > |         Method               | Instruction Tokens    | User Input Tokens  | Compress Ratio | CPU Latency |
> > > |------------------------|:---------:|:---------:|:---------:|:---------:|
> > > | Baseline     | 266 | 51 | 1 | 1398ms |
> > > |AutoCompress |50| 51 |3.14x| 821.4ms|
> > > |LoRA-Gen| 0 |51 |6.22x| 673.1ms|
> > >
> > > **Table 23:Efficiency Comparison.**
> > >
> > >
> > > **Q4: Questions on figure 1.**
> > >
> > > **Ans:**
> > > In Figure 1, we utilize the same few-shot samples across different test cases, allowing the cloud-side LM to perform a single inference to generate re-parameterized weights in the 1/3/5-shot setting, respectively. Therefore, the edge-side model (Qwen-1.5B) utilizes the same specialized parameters to complete this task evaluation without additional prefix prompts, which achieves a constant average inference time in Figure 1.
> > > Moreover, the latency of a single forward pass of the cloud-side model is negligible compared to the evaluation of the entire dataset.
> > > The paradigm of a single system prompt serving multiple user inputs is widely observed in real-world scenarios[1,2,3].
> > >
> > > We thank you for the precious review time and comments. Please let us know if you have any unsolved or other concerns.
> > >
> > > [1] Abdullahi, T., Singh, R. and Eickhoff, C., 2024. Learning to make rare and complex diagnoses with generative AI assistance: qualitative study of popular large language models. JMIR Medical Education, 10(1), p.e51391.
> > >
> > > [2] Wang, Z.M., Peng, Z., Que, H., Liu, J., Zhou, W., Wu, Y., Guo, H., Gan, R., Ni, Z., Yang, J. and Zhang, M., 2023. Rolellm: Benchmarking, eliciting, and enhancing role-playing abilities of large language models. arXiv preprint arXiv:2310.00746.
> > >
> > > [3] Xi, Z., Chen, W., Guo, X., He, W., Ding, Y., Hong, B., Zhang, M., Wang, J., Jin, S., Zhou, E. and Zheng, R., 2023. The rise and potential of large language model based agents: A survey. arXiv preprint arXiv:2309.07864.

---

> > > > ### Comment · Reviewer_WeXV · 2024-11-26
> > > > **On Statistical Testing and Comparisons to other Prompt Compression Techniques**
> > > >
> > > > Thanks for the quick reply! Let me be clear on my concern in these two items. Currently, the results in Table 20 and Table 21 show  the proposed method does not offer a statistically significant improvement over AutoCompress in performance and is on the order of a few milliseconds different in runtime.
> > > >
> > > > In your response you state "the improvements of our method in the LoRA-MoE-based approaches... are our core contributions". I agree with you that you have demonstrated improvements over LoRA-MoE-based approaches in terms of latency! However, improving LoRA-MoE is distinct from the high-level highlighted contributions you note in the introduction of your work on lines 75-93.
> > > >
> > > > I think the goals you lay out are well-stated and important, but along with them comes an expectation that you compare to the strongest related works that achieve similar goals.
> > > >
> > > > Let me lay out why I think AutoCompress is such a baseline:
> > > > 1) "Context compression for unseen tasks". AutoCompress also directly aims to tackle this, so they are comparable in this regard.
> > > > 2) "Reparameterized model... avoiding additional inference costs". The key metric for this claim is the inference time efficiency which, on GPU, appears nearly indistinguishable for AutoCompress.
> > > > 3) "our method does not require any additional training". AutoCompress also does not require additional training at inference time, only pretraining similar to LoRA-Gen, so they are comparable in this regard..
> > > > 4) "Knowledge Transfer...which enhances performance effectively" The key metric for this claim is the performance on your benchmarks, by which LoRA-Gen does not offer a statistically significant improvement over AutoCompress.
> > > >
> > > > Given that AutoCompress is older, more widely cited, and (by your own results) a stronger method than LoRA-MoE, it's unclear why improving over LoRA-MoE is important enough on it's own to be viewed as the core contribution.

---

> > > > > ### Comment · Reviewer_WeXV · 2024-11-26
> > > > > **On "Meta Tokens" (Cont)**
> > > > >
> > > > > Thanks for the updates on the method with respect to Meta tokens. The added paragraph from 196 to 203 make the method significantly clearer!
> > > > >
> > > > > I have updated my presentation score from 2->3 accordingly.

---

> ### Author Response · Authors · 2024-11-23
> **Rebuttal by authors 2/2**
>
> **Q6: Require a lot more explanation than the description on 216-220?**
>
> **Ans:**
> Following meta-learning, we seek to utilize a unified representation linked to task-specific information so as to improve the generalization capabilities across various tasks. Therefore, this representation is termed as meta-token, generated by the cloud-side large language model autoregressively.
> Specifically, For each given task description, we derive L tokens, one for each layer of the edge-side language model, with each meta-token directing expert routing at its respective layer, referring to lines 197 - 202 of our revision.
>
> **Q7: Confusion on Figure 1**
>
> **Ans:**
> It seems there exists some misunderstanding. All methods employ few-shot samples as the prefix input. Specifically, these samples are injected into the cloud-side large model in our method, while other approaches concatenate them directly with the user input for the edge-side model (Qwen-1.5B).
> Furthermore, we aim to illustrate our inference advantage, which is particularly relevant in application-driven scenarios.
>
> **Q8: Typo and Citation Error.**
>
> **Ans:**
> We sincerely apologize for the errors. The citation issues have been modified, including the correct LLaMA3 citation (please refer to lines 53, 321, and 475). The Typo error (``by generates") has been corrected in line 200. Thanks again.

---

> > ### Comment · Reviewer_WeXV · 2024-11-25
> > **On Figure 1**
> >
> > Thank you for your response! Unfortunately, it has raised another question for me. If "these samples are injected into the cloud-side large model in our method", how does this increase in the context length fed to the cloud-side large model have no impact on the latency metric for your method shown in Figure 1?
> >
> > For LoRA-Gen, the latency is shown as constant (vertical) in Figure 1 which seems impossible if there are an increased number of samples sent to the cloud side model for your method.
> >
> > Moreover, if your method requires running inference with both the cloud-side model and the client-side model, how is it **faster** than the Qwen 1.5B latency, which only requires running inference with the client-side model?

---

> ### Comment · Reviewer_WeXV · 2024-11-25
> **On "Meta-Tokens"**
>
> The term meta token still seems inadequately described in this response. Reviewer UJNj and Reviewer Kb9r also noted that this term confused them as one of the first items in their reviews, suggesting that this is a more general critique shared by multiple reviewers.
>
> **Concretely, is this statement from my original review accurate: "these tokens are not specialized for this parameter generation task but are just regular llama tokens given the prompt"?**
>
> At the current level of detail, that seems to be true based on your response "for each given task description, we derive L tokens" which are "generated by the cloud-side large language model autoregressively".
>
> Regardless, there are still concrete details that aren't clear:
> - Which output of the cloud-side large language model is used for the meta-token? Is it the final hidden state or the embedding of the discrete output token? If it's the embedding of the discrete output token, how is it sampled from the softmax distribution?
> - Since these $L$ tokens are generated autoregressively, how are earlier tokens incorporated into the context? Is this standard Llama decoding?
> - If the model is not trained to generate these output tokens, what is the rationale for assuming that each token corresponds to a layer? Without training, the generated tokens will represent a response to the task description prompt. If the model is trained to generate these output tokens, the method doesn't appear to be described.

---

> ### Author Response · Authors · 2024-11-28
> **Further response to Reviewer WeXV by authors on Statistical Testing and AutoCompressors Comparisons**
>
> We sincerely appreciate the reviewer's thoughtful and patient feedback. We completely understand your concerns and will further clarify the significant test and AutoCompressors comparison through two main points.
>
> 1. We realize that there is a technological mistake in the significance test of Table 21 (we mistakenly take the average accuracy result of the entire task as one sample, which means there are only 8 samples during the entire test, which is inconsistent with the test approach you provided [1]). To rectify this, we re-conduct the bootstrap test following the methodology outlined in [1]. Given that larger sample sizes enhance the reliability of bootstrap tests, our results on the Hellaswag dataset (10,046 test cases) are presented in Table 24.
>
> [1] https://aclanthology.org/P18-1128/
>
> |               | LoRAMoE    | MiXLoRA  | AutoCompressors |
> |------------------------|:---------:|:---------:|:---------:|
> | Bootstrap-Test   | [0.0063, 0.0338] | [0.0052, 0.0327] | [0.0022, 0.0297] |
>
> **Table 24: AutoCompressors with OPT-2.7b and LoRA-MoE-based methods with Gemma-2b.**
>
> 2. Our application scenario emphasizes small-sized language models on edge-side devices. To this end, we conduct an additional comparison with the AutoCompressors method using the OPT-2.7B model, as shown in Table 25.  Our method achieves a 1.5-point improvement in the average accuracy of unseen tasks over AutoCompressors. Additionally, the bootstrap test interval presented in Table 24 confirms a statistically significant enhancement compared to AutoCompressors. Furthermore, we measure the latency of AutoCompressors, with the results showing a 1.52x speedup that emphasizes the efficiency of our approach. The above outcomes align with the four contributions discussed in lines 75-93.
>
> |               | Hella    | WinoG  | PIQA | Average | Latency |
> |------------------------|:---------:|:---------:|:---------:|:---------:|:---------:|
> | AutoCompressors   | 44.7 |62.4 |73.3 | 60.1 | 11.4ms|
> | Ours  | 46.3 |63.7 |74.9 | 61.6 | 7.54ms|
>
>
> **Table 25: Comparison with AutoCompressors.**
>
> We fully agree with the reviewer that AutoCompressors should be considered as a baseline. We have added the comparison results with the AutoCompressors method to the appendix in the revision shown in lines 739-744. We thank you for the constructive feedback again and would like to confirm if this addresses the concerns regarding the significance test and the AutoCompressors baseline. We look forward to your kind response. Thanks a lot.

---

### Official Review · Reviewer_PsM9 · 2024-11-03

**Soundness:** 3
**Presentation:** 3
**Contribution:** 3
**Rating:** 6
**Confidence:** 3

**Summary:**

This work builds upon previous LoRA-based mixture-of-experts approaches for multi-task training in large language models. In classic LoRA-MoE methods, individual LoRA modules are fine-tuned within an LLM and selected using a routing function. Here, the authors propose an alternative method, consisting in generating a cloud-based LoRA module directly from a task-specific prompt. The generated module is then integrated into a general, edge-side model using reparameterization, creating a specialized LM adapted to the task at hand.

The authors show that this method offers equivalent or improved performance across a variety of tasks, and features additional improvements, mainly in the form of significant gains in inference speed and context length over previous methods. They verify their findings across several language models and multiple tasks.

**Strengths:**

The paper is well-written and clear. The proposed online LoRA-Gen method is interesting and innovative, and it achieves strong empirical results, namely in terms of inference speed and compression ratio gains over the previous methods. This is mostly due to efficient inference-time specialization without additional training, which is a significant benefit for deploying models on resource-constrained devices. In addition, LoRA-Gen offers good generalization to unseen tasks thanks to knowledge transfer from large to small models, resulting in a flexible and adaptable approach.

Given these points and the clear hyperparameter setup for reproducibility, I believe that the authors' method is likely to be useful in several practical applications.

**Weaknesses:**

My main point of criticism of this work is that its positioning with respect to previous methods seems a bit unclear and could use some improvement. In section 3.2, the method is described as addressing three challenges: effectiveness for multi-task learning, generalization to unseen tasks, and computational complexity. The results do indicate that LoRA-Gen performs well in all three aspects, but multi-task learning gains are quite modest compared to previous methods, especially given the added complexity of the indirect LoRA generation. It seems to me that the main benefit of LoRA-Gen is its impressive inference speedup and compression gains. As a result, the authors' findings would have more strength if claims of efficiency were discussed in more detail - for example, perhaps their method can allow edge-side optimizations for model memory usage, and other efficiency metrics such as computing costs and data requirements could be taken into account.

The section on limitations is very short and could be more detailed. The online component is the major strength of this method, but it also leads to cloud dependence. For example, the authors could describe possible use cases.

The paper contains a few minor formatting/typographical errors: "Sotmax" in Eq. 4, "which maintaining" in the "Intelligent Agent Scenario" subsection of 4.3, as well as issues with the formal of several citations in the first paragraph of 4.3, which should all be easy to correct.

**Questions:**

In the ablation study, it is said that the average accuracy decreases by 1.2 points when the auxiliary loss is excluded from training. I am assuming that this 1.2-point difference is computed from the accuracy obtained with a loss coefficient of 0.01, but please let me know if I am mistaken. This would imply that the model performance is worse with a 0.1 loss coefficient than if it has no auxiliary loss, and that it increases as the value of alpha decreases. Have the authors tried using even lower values to see whether model performance keeps increasing past the 0.01 point?

---

> ### Author Response · Authors · 2024-11-23
> **Rebuttal by authors**
>
> **Q1: Its positioning with respect to previous methods seems a bit unclear and could use some improvement.**
>
> **Ans:**
> Thanks for your valuable suggestions. We calculate the computing cost (FLOPs), GPU Memory and Latency across training and inference modes as shown in Table 12, and we have already presented it in the A.5 of our Appendix.
> MixLoRA† indicates the method without specific optimization. All metrics are measured on a Nvidia A100 GPU. FLOPs are measured using an input of 100 tokens and an instruction of 200 tokens, while memory and latency are evaluated in training mode with a batch size of 8 per GPU.
> By leveraging the unified representation from the aspect of meta-learning and reparameterization approach, we achieve minimal FLOPs and the shortest latency during the inference phase.
>
> | Method            | Training Mode FLOPs | Training Mode Memory | Training Mode Latency | Inference Mode FLOPs | Inference Mode Memory | Inference Mode Latency |
> |-------------------|---------------------|----------------------|-----------------------|----------------------|-----------------------|------------------------|
> | +LoRA             | 4.736E+11           | 37096MiB             | 0.85s                 | 4.708E+11            | 11208MiB              | 0.19s                  |
> | +LoRAMoE          | 4.742E+11           | 26326MiB             | 1.19s                 | 4.742E+11            | 11286MiB              | 0.22s                  |
> | +MixLoRA† | 5.061E+11           | 30844MiB             | 2.17s                 | 5.048E+11            | 11828MiB              | 1.08s                  |
> | +LoRA-Gen         | 1.667E+12           | 39603MiB             | 2.84s                 | 1.552E+11            | 10932MiB              | 0.11s                  |
>
> **Table 12: Efficiency Comparison.**
>
> **Q2: The section on limitations is very short and could be more detailed.**
>
> **Ans:**
> For offline scenarios involving fixed and extended system prompts, the cloud-side model can generate customized LoRA parameters in one inference, which are then supplied to the end-side model. We have revised it in the paper.
>
> **Q3: Minor formatting/typographical errors.**
>
> **Ans:**
> Sorry for the errors. We have revised them in Line 250 and Line 404 of the revision.
>
> **Q4: Have the authors tried using an even lower coefficient of auxiliary loss.**
>
> **Ans:**
> We have tried this hyperparameter tuning, detailed results are shown in Table 19. A coefficient of 0.001 is the best.
>
> | Loss Coefficient | ARC-c | ARC-e | OBQA  | SIQA  | WinoG | PIQA  | AVE. ↑ | HAR. ↑ |
> |------------------|-------|-------|-------|-------|-------|-------|--------|--------|
> | 0.1              | 41.3  | 73.1  | 31.8  | 54.5  | 65.9  | 75.9  | 57.1   | 51.7   |
> | 0.05             | 43.1  | 74.2  | 32.9  | 54.3  | 66.3  | 76.2  | 57.8   | 52.8   |
> | 0.005            | 41.6  | 72.8  | 28.8  | 54.5  | 66.9  | 76.3  | 56.8   | 50.5   |
> | w/o              | 39.8  | 71.5  | 29.0  | 53.0  | 66.2  | 76.8  | 56.1   | 49.8   |
> | 0.01             | 44.8  | 74.7  | 33.0  | 55.3  | 67.3  | 76.8  | 58.7   | 53.6   |
>
> **Table 19:Performance comparison under different loss coefficients.**

---

### Official Review · Reviewer_UJNj · 2024-11-05

**Soundness:** 3
**Presentation:** 3
**Contribution:** 3
**Rating:** 5
**Confidence:** 3

**Summary:**

**Summary**: The paper presents LoRA-Gen, a layerwise LoRA-MOE approach for specialized language models. The method employs a larger teacher LM (LLaMA-8B) to convert task-specific prompts into meta tokens, which are then used to generate adapter weights for a smaller target LM. The authors demonstrate improved accuracy and reduced latency compared to baselines like LoRA and LoRA-MOE.

**Detail**: This paper proposes to use a large teacher LM (llama-8b) to transform prompts (task definition, few-shot examples etc..) into meta tokens, and train a routing model to transform meta tokens into adaptor weights and finally assemble these adaptors with the corresponding weights to a target smaller LM. With this pipeline, long task specific prompts are compressed into LoRA, and the smaller LM can use these LoRA for downstream tasks.

**Results**: This paper conducted experiments with some reasoning classification tasks and agent task. Results show their method can significantly reduce the latency and get better results.

**Strengths:**

1. The method presents a novel approach to compress task-specific prompts into LoRA weights. And the layerwise LoRA-MOE design is an interesting architectural contribution to the field of model adaptation
2. The reduced latency could be valuable for real-world applications, particularly in resource-constrained settings
3. The evaluation includes both classification tasks and more complex agent-based scenarios

**Weaknesses:**

**Major Concerns**
1. Critical Implementation Details Missing

- The meta token generation process and their representation are not adequately explained
- The 'direct' method referenced in Table 8 lacks proper introduction
- The cloud-side LM's role during inference requires clarification
2. Questionable Latency Comparisons

The baseline methods are task-agnostic, which means they either support any task inference via prompt, or routing to task specific lora weights. But the proposed method is task specified, in my understanding, need to know which task it is processing to use its corresponding LoRA-gen weights. The latency advantages may primarily stem from task specification rather than architectural improvements. Please correct me if I'm wrong. Also, n line 259, it says "our method is cost-free during inference". I think it is not true when the testing is task agnostic.

3. Statistical Rigor Concerns

Results lack reporting of mean and variance metrics
This is particularly crucial given:

- The use of relatively small models (1.1-2B parameters)
- Small dataset sizes (e.g., WinoGrande)
- The potential variance in few-shot learning scenarios

**Minor issues**
1. The paper incorrectly groups diverse tasks under "Commonsense Reasoning Datasets." Suggest renaming to "Reasoning Tasks" as the datasets span both commonsense and scientific reasoning
2. Sections 4.1 and 4.3 contain redundant dataset introductions and citations
3. Dataset sizes should be explicitly stated for reproducibility

**Questions:**

N/A

---

> ### Author Response · Authors · 2024-11-23
> **Rebuttal by authors**
>
> **Q1: Critical Implementation Details Missing.**
>
> **Ans:**
> Thanks for your advice, we provide more explanation here and have revised our manuscript.
>
> 1. Please refer to the response of Reviewers Kb9r's Q2. We also have added a more detailed explanation in line 197.
>
> 2. Sorry for missing information. Specifically, both the "direct method" and the meta token (indirect way) are derived using the causal token paradigm of LLM and subsequently mapped to the parameter space via a feedforward neural network.
> The key difference lies in their shapes: the $i$-th token of the former has a shape of $[1, 3\times 2\times d\times r]$, whereas the meta token has a shape of $[1, n]$.
> $d$, $r$, and $n$ indicate the hidden dimension of edge-side small LM, the low rank of the LoRA setting, and the number of LoRA experts, respectively. We have revised it in Line 464.
>
> 3. As the reviewer mentioned in the strengths section (real-world applications), we aim to extend our approach to industrial scenarios, which typically feature a fixed specialized system prompt and varying user inputs. In such cases, the cloud-side large model performs the generation of customized LoRA weights through a one-time system prompt inference and supplies these weights to the edge-side small model.
>
> **Q2: Questionable Latency Comparison and Task Agnostic Discussion.**
>
> **Ans:**
> First, we consider there are some misunderstandings. Methods such as MixLoRA and LoRA-MoE, are token-wise MoE strategies, which means they need to rout experts for each token, which will undoubtedly cause more latency.
> These approaches claim that the MoE strategy is able to mitigate knowledge forgetting and enhance generalization capability.
> We want to emphasize the reparameterize ability of our method where the generated weights can merge into SLM seamlessly without compromising performance.
> Then, we propose a fresh perspective in this paper: employing a large cloud-side LM to generate parameters for a smaller edge-side model, enabling better specialization.
> As the reviewer mentioned, this strategy is highly relevant in industrial contexts, which are typically task-specific. In such cases, users provide a consistent system prompt but submit a wide variety of questions, which general methods fail to manage.
> Moreover, we consider that the generated LoRA may reluctantly handle task-agnostic scenarios given a general system prompt.
>
> **Q3: Statistical Rigor Concerns.**
>
> **Ans:**
> 1. We prioritize edge-side scenarios, where computing resources are constrained, leading us to concentrate on the effectiveness of small-size models.
> Furthermore, we also evaluate the performance of an 8B-parameter model (Llama3-8b) without additional system prompt, as outlined in Table 17:
>
> | Method     | ARC-c | ARC-e | OBQA  | BoolQ | SIQA  | HellaS | WinoG | PIQA  | AVE.  | HAR.  |
> |------------|-------|-------|-------|-------|-------|--------|-------|-------|-------|-------|
> | Baseline   | 53.2  | 81.6  | 34.2  | 83.2  | 52.6  | 57.7   | 71.6  | 78.6  | 64.1  | 59.1  |
> | + LoRA-Gen | 57.5  | 83.6  | 38.2  | 84.5  | 59.8  | 58.2   | 73.3  | 79.8  | 66.9  | 62.8  |
>
> **Table 17: Performance Comparison between our method and baseline based on Llama3-8B.**
>
> 2. We conduct the evaluation following counterparts such as MixLoRA and LoRAMoE. The testing data size of all tasks is summarized in Table 18:
>
> |                  | ARC-c | ARC-e | OBQA | BoolQ | SIQA | HellaS | WinoG | PIQA  |
> |------------------|-------|-------|------|-------|------|--------|-------|-------|
> | Number of samples| 1171  | 2380  | 500  | 3270  | 1954 | 10042  | 1267  | 1838  |
>
> **Table 18: Data size for each task evaluation.**
>
> 3. Across all experiments, we randomly select few-shot examples to ensure robustness. Additionally, for varying few-shot counts shown in Figure 1 of the manuscript, we calculate the results multiple times, resulting in a standard deviation of 0.0146.
>
> **Q4: Writing Questions about manuscript.**
>
> **Ans:**
> 1. We sincerely appreciate your suggestion and have updated the manuscript accordingly. Please refer to line 298 for the correct group name.
>
> 2. There seems to be some misunderstanding here. Section 4.1 provides an overview of the meta-information of the data used, while Section 4.3 details how we allocate the dataset into seen and unseen parts to evaluate our capabilities in multi-task learning and generalization to unseen tasks.
> However, we acknowledge that there are indeed duplications in the citation part. We have addressed this issue and made the necessary modifications (shown in line 363). Thanks again for the reviewer's valuable suggestion.
>
> 3. Thanks for your advice, we have added this information in A.4 of Appendix

---

> > ### Author Response · Authors · 2024-11-28
> > **Further discussion with Reviewer UJNj**
> >
> > Dear Reviewer UJNj,
> >
> > We thank you for the precious review time and comments. We have provided corresponding responses and results, which we believe have covered your concerns. We hope to further discuss with you whether or not your concerns have been addressed. Please let us know if you have any unsolved or other concerns and we look forward to your kind response.
> >
> > Thanks,
> >
> > Paper 6234 Authors.

---

> > ### Comment · Reviewer_UJNj · 2024-12-01
> > **Feedback of rebuttal -1**
> >
> > Thank you for your response. I appreciate the authors' additional experiments and clarifications. While some of my concerns have been addressed, others remain unresolved.
> >
> > - Regarding the missing information from the initial draft, I now understand what the 'direct method' is. However, could you confirm whether the generation of meta tokens is training-free and entirely controlled through prompting?
> > - The rebuttal hasn't changed my perspective on the task-agnostic versus task-specific nature of the baseline methods and the proposed method. While I acknowledge that token-level routing is slower than layer-wise routing, there's an important distinction: MixLoRA is designed for multitask solving, whereas LoRAGen is task-specific. I appreciate the acknowledgment added in the Limitations section. However, I remain concerned about the fairness of comparison between these two methods, particularly in Table 2.
> > - I appreciate the inclusion of results from a larger model. However, my initial concern about Table 2's results remains: they need mean and variance reporting. For instance, with Gemma-2B, LoRA-Gen achieves 63.9 while baselines range from 63.5 to 63.9. Given these small differences, multiple runs would help better understand the statistical significance of these results.
> >
> > Regarding the new experiments presented in this rebuttal:
> >
> > What is the 'baseline' referring to? Is it LoRA, LoRAMoE, or MixLoRA?
> > Why does LoRA-Gen show much greater improvement here compared to Table 2?

---

> ### Author Response · Authors · 2024-12-02
> **Further response to Reviewer UJNj by authors**
>
> We greatly appreciate your additional feedback. Our responses to each point are provided below:
>
> 1. As noted in line 321, generating meta-tokens involves a training phase.
>
> 2. We perform the multi-task evaluation as shown in Table 2, which is consistent with the original objective of MixLoRA. Focusing on the seen-task section, we train and evaluate across these five joint tasks, achieving a 5.3x speedup (MixLoRA's 141.9ms vs ours 26.7ms on Qwen-1.5B) with an average accuracy of 57%, compared to MixLoRA's 56%. Although our method excels in specific-task scenarios, this result underlines its comparable performance in multi-task cases.
>
> 3. We fully understand your concern. Due to time constraints, we take Gemma-2B you mentioned as an example and conduct two runs. The statistical mean and variance are presented in Table 26. We will update the results in the revised version.
>
> | Method     | ARC-c | ARC-e | OBQA  | BoolQ | SIQA  | HellaS | WinoG | PIQA  | AVE.  |
> |:------------:|:-------:|:-------:|:-------:|:-------:|:-------:|:--------:|:-------:|:-------:|:-------:|
> | LoRAMoE | 50.5&plusmn;0.57 | 81.9&plusmn;0.14 | 38.5&plusmn;0.42|78.3&plusmn;0.21|54.9&plusmn;0.42|53.8&plusmn;1.06|73.1&plusmn;0.28|79.2&plusmn;0.14|63.78&plusmn;0.07  |
> |MixLoRA|52.5&plusmn;0.21|79.8&plusmn;0.57|38.2&plusmn;0.57|75.5&plusmn;0.14|59&plusmn;0.21|54.1&plusmn;0.07|72.6&plusmn;0.13|78.6&plusmn;0.56|63.79&plusmn;0.03
> |LoRA-Gen|51.4&plusmn;0.21|81.7&plusmn;0.28|38.6&plusmn;0.57|76.8&plusmn;0.85|55.5&plusmn;0.14|56.1&plusmn;0.14|71.4&plusmn;0.28|79.6&plusmn;0.14|63.89&plusmn;0.01
>
> **Table 26: Mean and variance results.**
>
> 4. Baseline in Table 17 refers to the native LLaMA3-8b without finetuning.

---

### Official Review · Reviewer_Kb9r · 2024-11-12

**Soundness:** 3
**Presentation:** 3
**Contribution:** 4
**Rating:** 8
**Confidence:** 3

**Summary:**

Generic LLMs often demonstrate a tradeoff between efficiency and effectiveness for domain-specific tasks or preferences. Often, we utilize parameter-efficient finetuning techniques to train task or dataset specific models, among which LoRA tuning is a very popular approach.

In this paper, the authors propose LoRA-Gen which utilizes an online Cloud-side language model (a finetune LLM with LoRA experts) to generate meta tokens based on the task-specific system instructions; these tokens controls the composition of the parameters from the LoRa experts for the task-specific specialized language models.

The authors empirically demonstrate that LoRA-Gen leads to several advantages over previous parameter-efficient tuning methods: 1) the system instructions are used to learn the specialized LoRA parameters, achieving context compression for the user queries, 2) more efficient than LoRA-MOE, and 3) knowledge-transfer from large cloud LLMs to specialized LMs.

The proposed approach is evaluated on commonsense reasoning and agenic benchmarks. The results demonstrate the superiority of the approach over existing parameter-efficient finetuning techniques on the above mentioned points.

**Strengths:**

The proposed approach, LoRA-Gen, is a novel approach for parameter-efficient finetuning with multiple strong advantages (listed above) over previous methods. The authors strongly justify their claims with strong results and ablations on reasoning and agentic benchmarks.

**Weaknesses:**

Please explain “edge-side language models”. It is used throughout the paper without properly introducing it.

More explanations **with examples** are needed regarding how the meta tokens are generated and used to learn final LoRA parameters. How is each meta token associated to a transformer layer in the edge-side LM?

Add confidence intervals to your results in Table 2 and 3.

Citation issues:

Correct all citations. For example:
“in specific tasks Fu et al. (2023); Grangier et al. (2024); Shen et al. (2024)”  to “in specific tasks (Fu et al. 2023; Grangier et al. 2024; Shen et al. 2024)”
“ (e.g., LLaMA3 Touvron et al. (2023b))” to  (e.g., LLaMA3; Touvron et al. 2023b)”
“Unlike LoRA-MoE Dou et al. (2024)” to “Unlike LoRA-MoE (Dou et al. 2024)”

**Questions:**

“consistently outperforms other fine-tuning methods across different backbone models.” ->
LoRA-Gen performance on SIQA is inferior to other models. Also performance improvements for Gemma-2B models are not consistent.

---

> ### Author Response · Authors · 2024-11-23
> **Rebuttal by authors**
>
> **Q1: Please explain “edge-side language models”.**
>
> **Ans:**
> Thanks for the great advice, we have added the explanation on line 33 of the revision.
> More specifically, ``edge-side language model" is a term in the industry domain, that usually indicates a powerful artificial intelligence system deployed on edge devices, such as mobile phones and embedded systems. It operates independently to deliver efficient, real-time intelligent services and is typically optimized to minimize computational and storage costs.[1]
>
> [1] Qu, Guanqiao, et al. "Mobile edge intelligence for large language models: A contemporary survey." arXiv preprint arXiv:2407.18921 (2024).
>
> **Q2: More explanations about the meta tokens generation and utilization.**
>
> **Ans:**
> Referring to line 197 of the manuscript, Given a series of few-shot samples or task-specific system prompts as input of cloud-side LM, the LM appends $L$ special tokens called <$meta$> behind them and transfers the inherent knowledge into these tokens with causal masks in a single forward pass.
> We define these tokens as meta tokens $\\{T_i^{meta}\\}_ {i=1}^L$, where $L$ represents the number of layers of the subsequent edge-side small language model (SLM).
> We take the $i$-th layer of SLM in the edge-side device as an example to show how the $i$-th meta token indicates the generation of specialized LoRA weights.
> We first utilize a lightweight feedforward neural network (2 linear layers with Batch Normalization) to transform the last hidden state of the token to expert space and get the router $R^i$.
> Then we adopt a KeepTop-K strategy to obtain the gate $G^i$ of $n$ experts (in case of $\{K=2, n=3\}$, pseudo $G^i$ may be $[0.83, 0.17, 0 ]$ ).
> Finally, the reparameterized weight of $i$-th layer can be formulated as: $\bar{w}^i = w^i + \sum_{j=1}^{n}G  ^iE_j$.
>
> **Q3: Add confidence intervals to your results in Table 2 and 3.**
>
> **Ans:**
> The standard error is illustrated in Table 10, and the results are also provided in A.3 of the Appendix.
> | Method            | ARC-c   | ARC-e   | OBQA    | BoolQ   | SIQA    | HellaS  | WinoG   | PIQA    |
> |-------------------|---------|---------|---------|---------|---------|---------|---------|---------|
> | LoRA-Gen (Ours)   | 0.0146  | 0.0089  | 0.0219  | 0.0076  | 0.0112   | 0.0050  | 0.0134  | 0.0100  |
>
> **Table 10: Standard error on language model benchmarks.**
>
> **Q4: Citation issues.**
>
> **Ans:**
> Thanks for the valuable suggestions. We have addressed these issues in the revision.
>
> **Q5: Performance improvements are not consistent.**
>
> **Ans:**
> Our core contribution lies in providing a specialized edge-side model that combines strong generalization capabilities with context compression for unseen tasks, which balances effectiveness and efficiency across both seen and unseen tasks.
> Accordingly, we use the harmonic mean, arithmetic mean across various tasks, and latency to evaluate the overall advantages of our method compared to counterparts, rather than focusing on specific tasks.
> Apologies for any possible misunderstanding; the polished version is shown in line 365.

---

### Note · Authors · 2025-01-23

I have read and agree with the venue's withdrawal policy on behalf of myself and my co-authors.